# Multikinase Treatment of Glioblastoma: Evaluating the Rationale for Regorafenib

**DOI:** 10.3390/cancers17030375

**Published:** 2025-01-23

**Authors:** Ana Maria Muñoz-Mármol, Bárbara Meléndez, Ainhoa Hernandez, Carolina Sanz, Marta Domenech, Oriol Arpí-Llucia, Marta Gut, Anna Esteve, Anna Esteve-Codina, Genis Parra, Cristina Carrato, Iban Aldecoa, Mar Mallo, Estela Pineda, Francesc Alameda, Nuria de la Iglesia, Eva Martinez-Balibrea, Anna Martinez-Cardús, Anna Estival-Gonzalez, Carmen Balana

**Affiliations:** 1Pathology Department, Hospital Universitari Germans Trias i Pujol, 08916 Badalona, Spain; ammunoz.germanstrias@gencat.cat (A.M.M.-M.); carosanz.germanstrias@gencat.cat (C.S.); ccarrato.germanstrias@gencat.cat (C.C.); 2Molecular Pathology Research Unit, Hospital Universitario de Toledo, 45005 Toledo, Spain; bmelendez@sescam.jccm.es; 3Medical Oncology, Institut Catala d’Oncologia (ICO), 08916 Badalona, Spain; ahernandezg@iconcologia.net (A.H.); mdomenechv@iconcologia.net (M.D.); aesteve@iconcologia.net (A.E.); 4Badalona Applied Research Group in Oncology (B-ARGO Group), Institut Investigació Germans Trias i Pujol (IGTP), 08916 Badalona, Spain; amartinezc@igtp.cat; 5CARE Program, Germans Trias i Pujol Research Institute (IGTP), Ctra de Can Ruti, Cami de les Escoles s/n, 08916 Badalona, Spain; embalibrea@iconcologia.net; 6Cancer Research Program, Institut Hospital del Mar d’Investigacions Mèdiques (IMIM), 08003 Barcelona, Spain; oarpi@imim.es; 7Centro Nacional de Análisis Genómico, Universitat de Barcelona (UB), C/Baldiri Reixac 4, 08028 Barcelona, Spain; marta.gut@cnag.eu (M.G.); anna.esteve@cnag.eu (A.E.-C.); genis.parra@cnag.eu (G.P.); 8Department of Pathology, Biomedical Diagnostic Centre (CDB) and Neurological Tissue Bank of the Biobank-IDIBAPS, Hospital Clinic, University of Barcelona, 08036 Barcelona, Spain; ialdecoa@clinic.cat; 9Unidad de Microarrays, Institut de Recerca Contra la Leucèmia Josep Carreras (IJC), ICO-Hospital Germans Trias i Pujol, Universitat Autònoma de Barcelona, 08916 Badalona, Spain; mmallo@carrerasresearch.org; 10Medical Oncology, Hospital Clínic, Translational Genomics and Targeted Therapeutics in Solid Tumors, August Pi i Sunyer Biomedical Research Institute (IDIBAPS), 08036 Barcelona, Spain; epineda@clinic.cat; 11Pathology Department, Neuropathology Unit, Hospital del Mar, Institut Hospital del Mar d’Investigacions Mèdiques (IMIM), 08003 Barcelona, Spain; 11669faq@gmail.com; 12IrsiCaixa AIDS Research Institute, Hospital Universitari Germans Trias i Pujol, 08916 Badalona, Spain; ndelaiglesia@irsicaixa.es; 13ProCURE Program, Catalan Institute of Oncology, Ctra. de Can Ruti, Camí de les Escoles s/n, 08916 Badalona, Spain; 14Medical Oncology, Hospital Universitario Insular de Gran Canaria, 35016 Las Palmas de Gran Canaria, Spain; aestgong@gobiernodecanarias.org

**Keywords:** glioblastoma, multikinase treatment, regorafenib, molecular target

## Abstract

For a targeted drug to be effective, the target must be present. To determine if this condition was met for regorafenib in glioblastoma, we analyzed 46 genes encoding protein kinases (PKs) inhibited by regorafenib in preclinical studies. We further focused on a subset of 18 genes encoding PKs inhibited by regorafenib at clinically achievable concentrations, which—together with the anti-angiogenetic activity of regorafenib—constituted the rationale for the REGOMA and AGILE trials. Oncogenic/likely oncogenic mutations, gene amplification, fusions, and/or gene overexpression indicated a potential regorafenib target. Thirty-four (33%) and twenty-six (25.2%) patients harbored at least one such alteration in the 46- and 18-gene sets, respectively. However, these results may overestimate the effectiveness of regorafenib due to its difficulty in penetrating the blood–brain barrier to reach the target. Future use of multi-target drugs should be guided by a thorough understanding of target presence and the drug’s ability to reach the target.

## 1. Introduction

Glioblastoma (GBM), the highest-grade astrocytoma tumor, poses a formidable challenge. Despite initial treatment involving surgery, irradiation, and temozolomide, virtually all patients relapse, with no current curative options [1]. Progress in treatment has been hampered by various factors, including the complexities of assessing clinical benefit, determining trial endpoints, and overcoming the blood–brain barrier (BBB) to reach the tumor. Additionally, identifying the optimal points for intervention—whether molecular alterations, angiogenesis, or metabolism—remains a complex task. Current treatment approaches include chemotherapy, immunotherapy, targeted therapy, and anti-angiogenic approaches, and choosing the most effective approach can be complicated [2,3].

Nitrosoureas, especially lomustine, have emerged as the most widely accepted systemic treatment for recurrent GBM and constitute the control arm in most phase II–III comparative trials [3]. Bevacizumab, known for its potent anti-edema effects, is also an accepted treatment in several countries [2]. Targeted therapy may be an option for the approximately 3% of patients harboring the *BRAF* V600 mutation, who can be treated with blockade of the MAPK pathway using dabrafenib plus the MEK inhibitor trametinib [4]. Additionally, the 1–2% of patients with *NTRK* gene fusions could potentially respond to larotrectinib or entrectinib [5]. Various protein kinase (PK) inhibitors have also been explored in several small studies [6,7,8,9].

Regorafenib is an oral multikinase inhibitor that was initially developed as a RAF1 kinase inhibitor in the 1990s [10]. A series of clinical trials led to the approval of regorafenib for palliative treatment of several tumors, including metastatic colorectal cancer, metastatic gastrointestinal stromal tumors, and hepatocellular carcinoma [11,12,13]. During the early stages of regorafenib development, preclinical studies explored the potential inhibitory effects of regorafenib on a wide variety of PKs [14,15,16]. In biochemical and cellular assays, regorafenib—or its primary active metabolites, M-2 and M-5—demonstrated inhibitory effects at clinically achievable concentrations on 18 membrane-bound and intracellular PKs: VEGFR1-3, KIT, PDGFRA/B, FGFR1-2, TIE-2, DDR2, NTRK1, EPHA2, RAF1, BRAF, MAPK11, FRK, ABL1, and RET (https://go.drugbank.com/drugs/DB08896#BE0000029, accessed on 13 May 2023; https://pubchem.ncbi.nlm.nih.gov/compound/Regorafenib, accessed on 13 May 2023; https://www.accessdata.fda.gov/drugsatfda_docs/label/2013/204369lbl.pdf, accessed on 13 May 2023; https://www.ema.europa.eu/en/documents/product-information/stivarga-epar-product-information_en.pdf, accessed on 13 May 2023). These PKs are involved in both normal cellular functions and pathological processes, including oncogenesis, metastasis, and tumor immunity. Regorafenib also exerted an anti-angiogenesis effect in xenograft models [17].

Based on this broad spectrum of PK inhibition and on further evidence of activity in xenograft brain tumor models [18,19], regorafenib was explored in the phase II REGOMA trial [9], in which 119 recurrent GBM patients were randomized to receive either lomustine or regorafenib. A slight benefit was observed for regorafenib in median overall survival (5.6 vs. 7.4 months; HR 0.50, 95% CI 0.33–0.75; log-rank *p* = 0.0009) and in median progression-free survival (1.9 vs. 2.0 months; HR 0.65, 95% CI 0.45–0.95; log-rank *p* = 0.022). Toxicity and tolerance were similar to those reported in other studies conducted in different diseases [11,12,13]. These results led to the approval of regorafenib by the Italian Medicines Agency (AIFA) and its inclusion in 2020 in the National Comprehensive Cancer Network guidelines as a new treatment option for recurrent GBM. However, a subsequent trial by the AGILE adaptative platform, where regorafenib was included as one of the arms to validate the positive REGOMA results, did not confirm these results, and in fact, the trial was stopped for futility [20]. Furthermore, no biomarker has been identified that can consistently predict which GBM patients are likely to benefit from regorafenib [21,22,23].

The effectiveness of multikinase inhibition of PKs for the treatment of GBM remains unclear. The efficacy of targeted therapy hinges on four crucial factors: the presence of the target; the capacity of the agent to precisely block the target; the ability of the agent to reach the location of the target; and, ultimately, the effectiveness of the agent in stopping disease progression [24]. In the present study, we sought to determine whether regorafenib met these four criteria for an effective therapy in GBM. Using RNA sequencing (RNA-seq) and whole-exome sequencing (WES), we analyzed molecular alterations in genes encoding PKs, including variants/mutations, copy number variations (CNVs), fusions, and aberrant expression, in a cohort of newly diagnosed GBM patients. We specifically examined the presence of regorafenib targets in our patient cohort, focusing on PKs with preclinical data demonstrating inhibition or affinity for regorafenib, particularly those inhibited at clinically achievable concentrations, as well as on genes involved in angiogenesis.

## 2. Materials and Methods

### 2.1. Patients

The GLIOCAT project was a retrospective multicenter study that included 415 consecutive patients from six institutions, all of whom were diagnosed with GBM as per the 2016 WHO CNS tumor classification and uniformly treated with surgery followed by radiation therapy in conjunction with concurrent and adjuvant temozolomide. No patient received regorafenib. WES results were available for 103 patients, 71 of whom also had available RNA-seq results.

### 2.2. Selection of Target PKs

Preclinical findings on the effect of regorafenib and/or its metabolites on 46 PKs were reported by Zopf et al. [14] and Wilhelm et al. [15]. Zopf et al. explored the pharmacological activity and pharmacokinetics of regorafenib and its metabolites M-2 and M-5. In their in vitro experiments, they examined the dissociation constants (K_D_ values) for 38 PKs. The K_D_ value relates to the binding inhibition activity (the affinity between a ligand and a protein) for a particular experiment. The lower the K_D_ value (lower concentration), the higher the affinity of the antibody [14]. Wilhelm et al. gave their results as the half-maximal inhibitory concentration (IC50) for ten PKs and added five PKs with IC50 values below 100 nM without detailing data. IC50 indicates the concentration at which an agent is capable of inhibiting a biological process by 50% [15].

Taken together, the two studies [14,15] provided comprehensive preclinical affinity and inhibitory data on 46 PKs, which comprised the set of PKs included in the present study. In case of doubt, preclinical data for some PKs were reviewed at https://pubchem.ncbi.nlm.nih.gov/compound/Regorafenib#section=BioAssay-Results, accessed on 20 May 2023). Although the biological concept between K_D_ and IC50 values is slightly different, the authors established a limit of ≤100 to determine a potential positive target affinity or inhibition [14,15,25,26]. For the sake of simplification in the present study, K_D_ and IC50 values were merged and represented graphically, which enabled comparison with our findings on the frequency of molecular alterations in each gene in our dataset.

We investigated the potential effectiveness of regorafenib in GBM using a three-pronged approach. First, we analyzed alterations in the 46 genes encoding protein kinases (PKs) for which preclinical inhibitory data exist. Second, based on data from DrugBank, PubChem, the EMA, and the FDA, which indicate that regorafenib inhibits only 18 of these PKs at clinically achievable concentrations, we focused on alterations in this subset of 18 genes. Third, given regorafenib’s known anti-angiogenic properties, we conducted a functional enrichment analysis of the 46 genes to identify those involved in anti-angiogenesis pathways that may contribute to its anti-angiogenic effects.

### 2.3. Molecular Analyses

We analyzed tumor tissue samples to identify gene mutations, CNVs, gene fusions, and aberrant gene expression in genes encoding the 46 PKs identified by Zopf and Wilhelm [14,15]. The symbols for genes encoding these PKs were standardized according to the HUGO Gene Nomenclature Committee (https://www.genenames.org, accessed on 12 June 2023), and protein names were standardized according to UniProt terminology (https://www.uniprot.org/ accessed on 13 June 2023). Table 1 provides a comprehensive summary of the nomenclature used as well as any prior aliases or symbols. This step was taken to accommodate recent shifts in terminology aimed at standardizing the identification of PKs.

The functional enrichment analysis showed that of the 46 genes studied, 19 are involved in angiogenesis-related pathways: *ABL1*, *BRAF*, *EPHA2*, *FGFR1*, *FGFR2*, *FLT1*, *FLT3*, *FLT4*, *KDR*, *KIT*, *MAPK9*, *MAPK11*, *MAP2K5*, *MAPK14*, *PDGFRA*, *PDGFRB*, *RAF1*, *TEK*, and *TIE1* (Table 1, Appendix A and Appendix A). Five of these nineteen genes are not among the eighteen that encode PKs targetable by regorafenib at clinically achievable concentrations (*FLT3*, *MAP2K5*, *MAPK9*, *MAPK14*, and *TIE1*), and there are no data on whether they can be targeted at clinically achievable concentrations of regorafenib. However, since the activity of regorafenib is also based on its anti-angiogenic effect, we categorized a group of 23 genes (18 with regorafenib inhibition of PKs and 5 with anti-angiogenic activity but no demonstrated inhibitory effect).

RNA-seq and WES were performed in conjunction with the GLIOCAT project as previously described [27,28,29,30,31]. For the present study, we re-analyzed our WES data to determine gene mutations (including single-nucleotide variants and InDels) and CNVs. Somatic variants identified by Mutect2 and processed as previously described [31] were functionally annotated (high and moderate impact) using SnpEff (https://pcingola.github.io/SnpEff/snpeff/inputoutput/#impact-prediction, accessed on 30 May 2023), with gene annotations sourced from ENSEMBL (version 75). We then classified the mutations according to their oncogenic potential based on the recommendations of Clinical Genome Resource (ClinGen), Cancer Genomics Consortium (CGC), and Variant Interpretation for Cancer Consortium (VICC) [32]. The CNV gain was analyzed in the same set of genes. To identify CNVs that could indicate gene amplification, we first compared copy numbers with RNA-seq gene expression findings but found no correlation (Appendix A). We then based our interpretation of CNVs on findings by French et al. [33], which suggest that EGFR amplification can reliably be determined by CNVs > 5 identified by next-generation sequencing (NGS). Hence, we accepted CNVs > 5 as a surrogate for gene amplification.

We re-analyzed our RNA-seq data to identify gene fusions and aberrant gene expression. We reviewed findings in tumor samples from the same patients [30] to identify fusions involving the target genes. We analyzed the expression levels for each gene in order to detect outlier samples with overexpression. The expression value (log2CPM) was considered an outlier based on the rule of 1.5 times the interquartile range above the third quartile.

Putative oncogenic alterations were defined as oncogenic/likely oncogenic mutations, gene amplification, gene fusions, and/or gene overexpression.

Finally, we performed a functional enrichment analysis (https://biit.cs.ut.ee/gprofiler/gost, accessed on 20 January 2024) of all 46 genes to identify those involved in angiogenesis, with the aim of exploring the potential anti-angiogenic effects of regorafenib [34].

### 2.4. Statistical Analysis

Patient data were double-anonymized before proceeding with the molecular studies. The frequency of mutations and CNVs was analyzed with R v3.4.2 and SPSS v24 and by manual examination. RNA-seq and WES findings are presented as absolute numbers with the corresponding percentages.

## 3. Results

The characteristics of the patients included in this study are detailed in Appendix A. Histological review confirmed a diagnosis of glioblastoma based on the WHO 2016 classification. According to the updated WHO 2021 classification, three patients would now be categorized as grade 4 astrocytomas due to the presence of an *IDH1* mutation. None of the patients exhibited alterations in *H3.3* genes. All participants underwent radiation therapy combined with concomitant and adjuvant temozolomide treatment.

Figure 1 shows the alterations detected in our series of patients in all 46 genes encoding the PKs on which preclinical assays showed an inhibitory effect of regorafenib [14,15]. However, subsequent in vivo assays detected regorafenib inhibition of only 18 of these 46 PKs at clinically achievable concentrations (Table 1).

### 3.1. Molecular Alterations in Genes

In all 46 genes, we analyzed gene mutations, amplification, overexpression, and fusions. Figure 1 summarizes the results of these analyses. Putative oncogenic alterations were defined as all oncogenic or likely oncogenic mutations (but not benign, likely benign, variants of uncertain significance (VUS), or VUS–nonsense mediated decay (VUS-NMD) mutations), amplification, overexpression, or fusions.

In 35 of the 46 genes, we detected a total of 87 canonical protein-coding mutations, all of which had the potential to influence their receptors by either moderate impact (a non-disruptive variant that might change protein effectiveness) or high impact (a variant that will probably cause protein truncation or loss of function or trigger nonsense-mediated mRNA decay) (Appendix A). The mutations were found most frequently in the following genes: *FLT4* (n= 7, 15.2%), *TEK* (*n* = 6, 13.0%), *KDR* (*n* = 5, 10.8%), *MAP3K19* (*n* = 5, 10.8%), *BRAF* (*n* = 4, 8.7%), *DDR2* (*n* = 4, 8.7%), *MKNK2* (*n* = 4, 8.7%), and *NTRK1* (*n* = 4, 8.7%). When the 87 mutations were classified according to their oncogenic potential [32], only 5 were classified as oncogenic/likely oncogenic: *BRAF* (*n* = 2, 4.3%), *PDGFRA* (*n* = 1, 2.1%), *DDR2* (*n* = 1, 2.1%), and *FGFR2* (*n* = 1, 2.1%) (Figure 1).

Fifteen of the forty-six genes (32.6%) had CNVs > 5, indicating amplification. The most frequent were *PDGFRA* (*n* = 9, 19.5%), *KIT* (*n* = 5, 10.8%), *KDR* (*n* = 3, 6.5%), *MAPK9* (*n* = 3, 6.5%), and *PDGFRB* (*n* = 2, 4.3%). Only one gene (*ABL1*) was part of a fusion (*ABL1*::*SZRD1*). Ten genes were overexpressed. *FRK* and *PDGFRA* were the most frequently overexpressed genes (Figure 1).

The genes with the highest frequency of any alterations were *PDGFRA* with fifteen alterations (32.6%), *KDR* with eight (17.3%), and *FLT4* and *KIT* with seven each (15.2%). Additionally, *EPHA2*, *FRK*, *MAPK9*, *MKNK2*, and *TEK* showed six alterations (13.0%) each. When only putative oncogenic alterations were considered, *PDGFRA* had the highest number of alterations (*n* = 14; 30.4%).

In summary, of the 46 genes studied, 40 had at least one alteration, while 6 (*CDK19*, *CDKL2*, *MAP3K20*, *NTRK3*, *TAOK3*, and *TNNI3K*) showed no alterations. When the analysis was limited to include only putative oncogenic alterations, only 22 genes had alterations, while 24 did not (Figure 1).

### 3.2. Identification of Potential Candidates for Regorafenib Treatment

#### 3.2.1. Alterations in Patient Samples

Patients could have more than one alteration in their tumor samples (Table 2 and Figure 1).

When we considered all alterations except benign or likely benign mutations, 73 patient samples (70.8%) had at least one alteration in one or more of the 46 genes studied, while 30 (29.1%) had none. Mutations were detected in 56 (54.4%), amplification in 17 (16.5%), a fusion in 1, and overexpression in 17 (16.0%) (Figure 1). In the subset of 18 genes encoding PKs targetable by regorafenib at clinically achievable concentrations, 48 patients (46.6%) had at least one alteration. In the expanded subset of 23 genes (the 18 plus the 5 angiogenesis-related genes not included in the 18-gene subset), 48 patient samples (46.6%) harbored alterations.

When we considered only putative oncogenic alterations, only 34 patients (33.0%) had at least one alteration in one or more of the 46 genes. Oncogenic/likely oncogenic mutations were detected in 5 (4.8%), amplification in 17 (16.5%), a fusion in 1, and overexpression in 17 (16.5%). In the subset of 18 genes encoding PKs targetable at clinically achievable concentrations, putative oncogenic alterations were detected in only 26 (25.2%). Finally, in the 23-gene subset, 30 patients (29.1%) had putative oncogenic alterations (Table 2 and Figure 1).

#### 3.2.2. Analysis of Potential Targets

Figure 2 shows the results of our comprehensive comparative analysis of 46 genes encoding PKs, displaying the affinity and inhibitory data provided by Zopf et al. [14] and Wilhelm et al. [15] and the molecular alterations that we detected in our patient samples. Our findings in the patients whose tumors harbored molecular alterations suggest an inhibitory potential of regorafenib or one of its metabolites on 42 of the 46 PKs. Zopf et al.’s data [14] provide insight into the receptor affinity of 39 PKs of the 46 genes studied. They found that regorafenib had affinity with 31 PKs. Additionally, its metabolites M-2 and M-5 had the potential to inhibit seven more PKs with a K_D_ ≤ 100. Unfortunately, no data were available for FGFR1-4, EPHA2, TEK, and NTRK1. Wilhelm et al.’s data [15] included information on FGFR1, EPHA2, and TEK but not on FGFR2-4 and NTRK1. Although FGFR1 (IC50 = 202) and TEK (IC50 = 311) were not found to be effectively targeted by regorafenib [15], consultation of PubChem revealed that subsequent assays confirmed the inhibition of FGFR1 and TEK but not that of NTRK1 and FGFR2-4. Therefore, tumors with alterations only in these genes would not be correctly inhibited. Nevertheless, only one of our patient samples had alterations only in these genes but also had overexpression of *RAF1*, indicating that it harbored a possible regorafenib target.

In summary, we performed three separate analyses to determine the percentage of our 103 patients who could potentially benefit from regorafenib treatment, based on the presence of putative oncogenic alterations and/or the anti-angiogenic activity of regorafenib (Table 3). The first analysis included all 46 genes encoding PKs identified by Zopf et al. [14] and Wilhelm et al. [15] and revealed that only 34 (33%) had putative oncogenic alterations and could thus be potential beneficiaries of regorafenib. In the second analysis, which focused exclusively on the 18 genes encoding PKs inhibited by regorafenib at clinically achievable concentrations, only 26 patients (25.2%) had putative oncogenic alterations and could potentially benefit from regorafenib. The third analysis expanded on the second analysis to account for the potential anti-angiogenic effect of regorafenib and added the 5 genes involved in angiogenesis pathways but not included in the 18-gene set of the second analysis. In this expanded group, only 30 (29.1%) had putative oncogenic alterations (Table 3).

Figure 3 displays a summary of our findings.

## 4. Discussion

Regorafenib efficacy in GBM has been explored in two randomized trials. The phase II REGOMA trial showed a modest survival benefit for regorafenib, but this finding was not confirmed by the GBM AGILE trial, a phase III Bayesian adaptive platform trial that tested multiple arms against a common control. In the GBM AGILE trial, regorafenib was tested in newly diagnosed GBM patients without *MGMT* promoter methylation, who were randomized against standard treatment with temozolomide, and in recurrent GBM patients, who were randomized against treatment with lomustine. The trial accrual was stopped for futility, and regorafenib activity was ruled out in both settings [9,20].

Targeted therapy for cancer relies on drugs that target specific molecules critical for cancer development and tumor growth. At present, no biomarker can identify GBM patients likely to benefit from targeted therapy with regorafenib [21,22,23]. In the present study, we explored the rationale behind using regorafenib, a multi-target drug, based on two factors that are crucial to the evaluation of a targeted therapy: GBM molecular alterations, indicative of the presence of the target, and data on preclinical regorafenib activity, indicating the capacity of regorafenib to block the target.

Regorafenib’s spectrum of action includes both 18 PKs targetable at clinically achievable concentrations and its anti-angiogenic capacity, which constituted the rationale for its selection as treatment for GBM in the REGOMA trial [9]. However, preclinical data indicate that regorafenib targets 46 PKs. When we assessed alterations in the 46 genes, we found that only 40 genes harbored alterations, which were present in 70.8% of our patients, although putative oncogenic alterations were present in only 33%. When we limited our analyses to the 18 PKs, although all 18 genes had alterations, they were present in only 46.6% of our patients, and putative oncogenic alterations were present in only 25.2%. Even when alterations in angiogenesis-related genes were included in our analysis, only 29.1% of patients harbored alterations that were potentially targetable by regorafenib. In theory, these would be the patients who could benefit from treatment with regorafenib, considering the different scenarios and assumptions.

Nevertheless, these data should be interpreted with caution, as the numbers may decline due to various factors. A comprehensive analysis revealed that 4 of the 46 PKs were not favorable targets for regorafenib (NTRK1, FGFR2, FGFR3, and FGFR4), and preclinical data for several PKs (EPHA2, FGFR1, and TEK) indicated that the concentrations of regorafenib needed for effective inhibition were quite high. In addition, data on inhibition or affinity obtained from preclinical studies cannot always translate to effective inhibition in cellular assays or in vivo studies [35]. Even if IC50 predictions are favorable, the translation to the clinical setting can be less than optimal, as drug activity can be affected by other factors, including protein binding, cell influx/efflux, absorption, metabolism, distribution to target compartments, transport and excretion, as well as BBB permeability [25,26,36], which is especially crucial in the treatment of brain tumors. Regorafenib and its metabolites exhibit high protein binding (99.5%), but BBB penetration is restricted by breast cancer resistance protein (BCRP/ABCG2) and P-glycoprotein (P-GP/ABCB1) [37]. In silico models predict poor BBB penetration of regorafenib [38], and studies have found that regorafenib levels in cerebrospinal fluid (CSF) are significantly lower than in serum. Levels of regorafenib and its metabolites were higher in serum than in CSF by a factor of 4–540 [39], with only 60% availability in pediatric patients [40]. Achieving effective tumor concentrations of regorafenib in the brain is thus unlikely at standard doses, and underdosing of the drug could compromise the inhibition of several PKs, especially those with less favorable preclinical data. These factors would further decrease the percentage of patients who would be potential candidates for regorafenib treatment even if the target were present in the tumor.

Our work has several limitations. Firstly, we contemplated any molecular alteration (other than methylation profile and histone modifications) as a potential cause of PK changes [41,42,43,44]. Although this approach may have oversimplified the underlying pathogenic processes, it was intentionally designed to include all possibilities and collect comprehensive data. However, many of the mutations we detected were VUS, and although we lack detailed knowledge about their effects on tumor development, PKs, and treatment response, they may well be non-driver mutations. In addition, different mutations in the same gene are not always effectively inhibited by the same targeted therapy, as has been observed in gastrointestinal stromal tumors with *KIT* mutations [45] and in lung cancer with *EGFR* mutations [46]. Furthermore, different alterations in a gene can cause different responses to a given treatment, as occurs with erdafitinib in bladder cancer [47]. Secondly, we determined the presence of gene amplification based on CNVs, where CNVs > 5 indicated gene amplification, but this approach, derived from findings on *EGFR* [33], may not apply to all the genes included in our study. Comparative data between FISH amplification and NGS results for these genes are lacking. Thirdly, our study did not explore the complex interactions between signaling pathways, cytogenetic clonal evolution, activation of alternative pathways, and other biological processes, such as the effects of the tumor microenvironment, autophagy, stem cells, synthetic lethality, or the de-inhibition of negative signaling feedback loops [48]. These factors, which are often considered indirect effects of tyrosine kinase inhibitors [49], cannot be accurately measured by molecular studies, and we currently lack a method to use them in selecting candidate patients for treatment. Finally, our study included only newly diagnosed patients. Recurrent patients may acquire new alterations differing from those at diagnosis. However, studying these alterations is complex due to the difficulty of obtaining new tissue samples at relapse [50].

Our findings raise critical questions about the future use of multi-target drugs in GBM without prior comprehensive knowledge of target presence, effective inhibition, and the drug’s ability to reach the brain at adequate concentrations. A more precise and targeted approach in drug development and clinical trials is necessary to improve success rates and avoid investing resources in treatments with uncertain benefits. Moving forward, a deeper understanding of the molecular landscape of GBM, especially at relapse, and advances in identifying predictive biomarkers could enhance the selection of patients likely to benefit from specific targeted therapies. Such an approach, based on greater comprehensive knowledge, will hopefully lead to more successful clinical trials and improved outcomes for GBM patients.

## Figures and Tables

**Figure 1 cancers-17-00375-f001:**
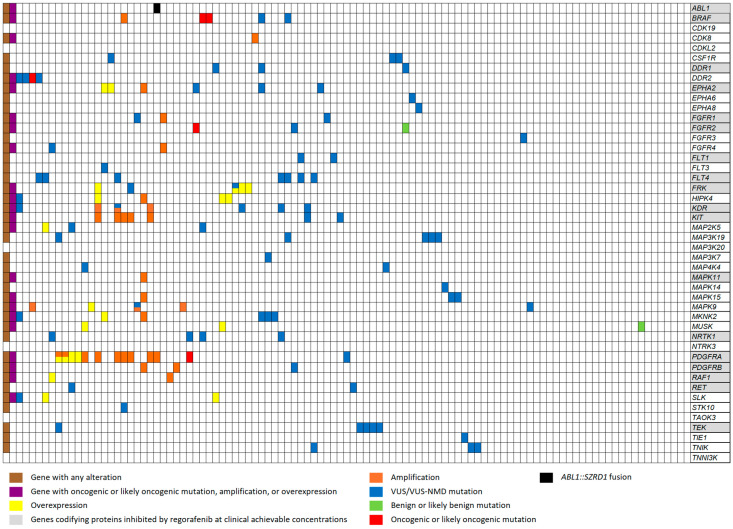
Molecular alterations found in our series of patients for the 46 genes encoding PKs potentially targetable by regorafenib.

**Figure 2 cancers-17-00375-f002:**
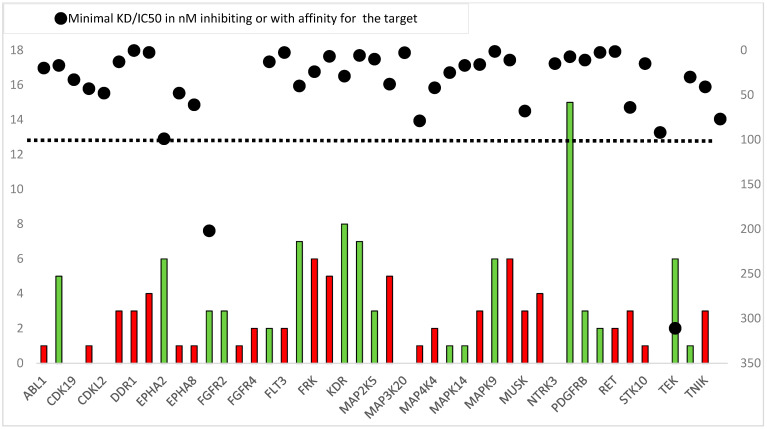
Molecular alterations detected in the 46 genes included in our study compared with the minimal regorafenib concentration (in nM) necessary for inhibition or affinity. Data for K_D_ and IC50 have been merged to simplify the figure. Columns show the number of alterations detected in each gene (left *y* axis). Dots show K_D_/IC50 in nM (right *y* axis). Green columns indicate genes involved in angiogenesis pathways. Dashed line represents the limit of ≤100 to determine a potential positive target affinity or inhibition.

**Figure 3 cancers-17-00375-f003:**
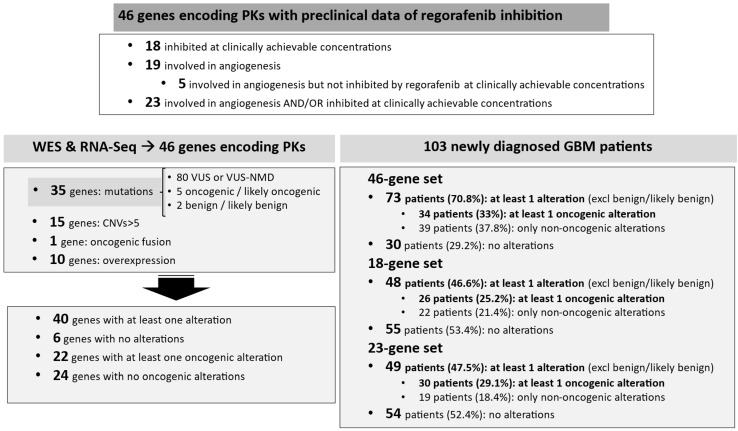
Study design and summary of results. We analyzed WES and RNA-seq data for tumor samples from 103 glioblastoma patients. We assessed mutations, amplifications, copy number variants (CNV) (as a surrogate for gene amplification), and expression in 46 genes encoding protein kinases (PKs) potentially inhibited by regorafenib. We then focused on 18 genes encoding PKs inhibited by regorafenib at clinically achievable concentrations. In addition, we performed a functional enrichment analysis of the 46 genes and found that 19 are involved in angiogenesis, 5 of which are not included in the 18-gene set. Putative oncogenic alterations comprised oncogenic/likely oncogenic mutations, gene amplification, and/or gene overexpression. Data in patients are shown for the 46-gene set, the 18-gene set, and a 23-gene set (comprising the 18 genes plus the 5 additional genes involved in angiogenesis). Results were calculated based on the presence of any alterations excluding benign or likely benign mutations) and on only putative oncogenic alterations.

**Table 1 cancers-17-00375-t001:** The 46 genes and corresponding PKs included in our study according to data obtained from Zopf and Wilhelm [14,15], showing the HUGO Gene Nomenclature Committee and UniProt terminology.

HUGO Term	PK Name	Previous PK Term	UniProt PK Term	PK Inhibited at Clinically Achievable Concentrations (N = 18) ^a^	Gene Involved in Angiogenesis Pathways (N = 19) ^b^
** *ABL1* **	Tyrosine-protein kinase ABL1	ABL1	ABL1	Yes	Yes
** *BRAF* **	Serine/threonine-protein kinase B-raf	BRAF	BRAF	Yes	Yes
** *CDK19* **	Cyclin-dependent kinase 19	CDK11	CDK19		
** *CDK8* **	Cyclin-dependent kinase 8	CDK8	CDK8		
** *CDKL2* **	Cyclin-dependent kinase-like 2	CDKL2	CDKL2		
** *CSF1R* **	Macrophage colony-stimulating factor 1 receptor	CSF1R	CSF1R		
** *DDR1* **	Epithelial discoidin domain-containing receptor 1	DDR1	DDR1		
** *DDR2* **	Discoidin domain-containing receptor 2	DDR2	DDR2	Yes	
** *EPHA2* **	Ephrin type-A receptor 2	EPHA2	ECK	Yes	Yes
** *EPHA6* **	Ephrin type-A receptor 6	EPHA6	EPHA6		
** *EPHA8* **	Ephrin type-A receptor 8	EPHA8	EPHA8		
** *FGFR1* **	Fibroblast growth factor receptor 1	FGFR1	FGFR1	Yes	Yes
** *FGFR2* **	Fibroblast growth factor receptor 2	FGFR1	FGFR2	Yes	Yes
** *FGFR3* **	Fibroblast growth factor receptor 3	FGFR3	FGFR3		
** *FGFR4* **	Fibroblast growth factor receptor 4	FGFR4	FGFR4		
** *FLT1* **	Vascular endothelial growth factor receptor 1	VEGFR1	VEGFR1	Yes	Yes
** *FLT3* **	Receptor-type tyrosine-protein kinase FLT3	FLT3	FLT3		Yes
** *FLT4* **	Vascular endothelial growth factor receptor 3	VEGFR3	VEGFR3	Yes	Yes
** *FRK* **	Tyrosine-protein kinase FRK	FRK/PTK5	FRK	Yes	
** *HIPK4* **	Homeodomain-interacting protein kinase 4	HIPK4	HIPK4		
** *KDR* **	Vascular endothelial growth factor receptor 2	VEGFR2	VEGFR2	Yes	Yes
** *KIT* **	Mast/stem cell growth factor receptor Kit	KIT	KIT	Yes	Yes
** *MAP2K5* **	Dual-specificity mitogen-activated protein kinase kinase 5	MEK5	MAP2K5		Yes
** *MAP3K19* **	Mitogen-activated protein kinase kinase kinase 19	YSK4	M3K19		
** *MAP3K20* **	Mitogen-activated protein kinase kinase kinase 20	ZAK	ZAK		
** *MAP3K7* **	Mitogen-activated protein kinase kinase kinase 7	TAK1	TAK1		
** *MAP4K4* **	Mitogen-activated protein kinase kinase kinase kinase 4	MAP4K4	M4K4		
** *MAPK11* **	Mitogen-activated protein kinase 11	p38-beta/SAPK2	MAPK11	Yes	Yes
** *MAPK14* **	Mitogen-activated protein kinase 14	p38-alpha	MK14		Yes
** *MAPK15* **	Mitogen-activated protein kinase 15	ERK8	MK15		
** *MAPK9* **	Mitogen-activated protein kinase 9	JNK2	JNK2		Yes
** *MKNK2* **	MAP kinase-interacting serine/threonine-protein kinase 2	MKNK2	MKNK2		
** *MUSK* **	Muscle, skeletal receptor tyrosine-protein kinase	MUSK	MUSK		
** *NTRK1* **	High-affinity nerve growth factor receptor	TRKA	NTRK1	Yes	
** *NTRK3* **	NT-3 growth factor receptor	TRKC	TRKC		
** *PDGFRA* **	Platelet-derived growth factor receptor alpha	PGFRA	PGFRA	Yes	Yes
** *PDGFRB* **	Platelet-derived growth factor receptor beta	PGFRB	PGFRB	Yes	Yes
** *RAF1* **	RAF proto-oncogene serine/threonine-protein kinase	RAF1	RAF1	Yes	Yes
** *RET* **	Proto-oncogene tyrosine-protein kinase receptor Ret	RET	RET	Yes	
** *SLK* **	STE20-like serine/threonine-protein kinase	SLK	SLK		
** *STK10* **	Serine/threonine-protein kinase 10	LOK	STK10		
** *TAOK3* **	Serine/threonine-protein kinase TAO3	TAOK3	TAOK3		
** *TEK* **	Angiopoietin-1 receptor	TIE2	TIE2	Yes	Yes
** *TIE1* **	Tyrosine-protein kinase receptor Tie-1	TIE1	TIE1		Yes
** *TNIK* **	TRAF2 and NCK interacting kinase	TNIK	TNIK		
** *TNNI3K* **	Serine/threonine-protein kinase TNNI3K	TNNI3K	TNI3K		

^a^ According to DrugBank, PubChem, the EMA and the FDA. ^b^ According to our functional enrichment analysis of the 46 genes. The genes encoding proteins inhibited by regorafenib at clinically achievable concentrations are shaded.

**Table 2 cancers-17-00375-t002:** Molecular alterations detected in patient samples. Each row represents one patient. A total of 46 genes were analyzed for the following alterations: mutations, fusions, amplification, and overexpression. Putative oncogenic alterations were oncogenic/likely oncogenic mutations, gene fusions, gene amplification, and/or gene overexpression. The presence of putative oncogenic alterations was assessed in three sets of genes: the extended 46-gene set (based on studies by Zopf et al. [14] and by Wilhelm et al. [15]); the limited 18-gene set (based on data from DrugBank, PubChem, the EMA, and the FDA: *ABL1*, *BRAF*, *DDR2*, *EPHA2*, *FGFR1*, *FGFR2*, *FLT1*, *FLT4*, *FRK*, *KDR*, *KIT*, *MAPK11*, *NTRK1*, *PDGFRA*, *PDGFRB*, *RAF1*, *RET*, and *TEK*); and a 23-gene set comprised of the 18 genes plus 5 angiogenesis-related genes not included in the 18-gene set (*FLT3*, *MAPK9*, *MAP2K5*, *MAPK14*, and *TIE1*). Table 3 shows the probability of the presence of potential regorafenib targets in patients.

Alterations Detected in Each Patient Sample ^a^ (46-Gene Set)	N alterations in Each Patient Sample (46-Gene Set)	Putative Oncogenic Alterations Detected in Each Patient Sample (46-Gene Set)	Putative Regorafenib Target? (46-Gene Set)	Putative Regorafenib Target? (18-Gene Set)	Putative Regorafenib Target? (23-Gene Set)
EPHA2 amp and MAPK11 amp and PDGFRB amp and MAPK15 amp and FLT3 amp and HIPK4 amp	6	EPHA2 amp and MAPK11 amp and PDGFRB amp and MAPK15 amp and FLT3 amp and HIPK4 amp	yes	yes	yes
KDR amp and KIT amp and PDGFRA amp and FRK over and HIPK4 over	5	KDR amp and KIT amp and PDGFRA amp and FRK over and HIPK4 over	yes	yes	yes
KDR mut and FLT4 mut and KDR amp and PDGFRA amp and KIT amp	5	KDR amp and PDGFRA amp and KIT amp	yes	yes	yes
KDR mut and DDR2 mut and HIPK4 mut and SLK mut and MKNK2 mut	5	-	no	no	no
BRAF amp and KIT amp and PDGFRA amp and STK10 mut	4	BRAF amp and KIT amp and PDGFRA amp	yes	yes	yes
PDGFRA amp and MAP3K19 mut (VUS-NMD) and TEK mut and PDGFRA over	4	PDGFRA amp and PDGFRA over	yes	yes	yes
MKNK2 mut and EPHA2 mut and DDR1 mut and BRAF mut	4	-	no	no	no
KDR amp and KIT amp and PDGFRA amp	3	KDR amp and KIT amp and PDGFRA amp	yes	yes	yes
FRK mut and PDGFRA amp and KIT amp	3	PDGFRA amp and KIT amp	yes	yes	yes
PDGFRA amp and MAP4K4 mut and MUSK over	3	PDGFRA amp and MUSK over	yes	yes	yes
FLT3 mut and EPHA2 over and MKNK2 over	3	EPHA2 over and MKNK2 over	yes	yes	yes
NTRK1 mut and FGFR4 mut and RAF1 over	3	RAF1 over	yes	yes	yes
RET mut and MAP2K5 mut and PDGFRA over	3	PDGFRA over	yes	yes	yes
NTRK1 mut and BRAF mut and MAP2K5 mut (VUS-NMD)	3	BRAF mut	yes	yes	yes
FLT4 mut and SLK over and MAP2K5 over	3	SLK over and MAP2K5 over	yes	no	yes
FGFR1 mut, MAPK9 mut and MAPK9 amp	3	MAPK9 amp	yes	no	yes
NTRK1 mut and FLT4 mut and KDR mut	3	-	no	no	no
FLT4 mut and BRAF mut and MAP3K19 mut (VUS-NMD)	3	-	no	no	no
PDGFRA amp and PDGFRA over	2	PDGFRA amp and PDGFRA over	yes	yes	yes
PDGFRA amp and ABL1::SZRD1 fusion	2	PDGFRA amp and ABL1::SZRD1 fusion	yes	yes	yes
MAPK9 amp and DDR2 mut	2	MAPK9 amp and DDR2 mut	yes	yes	yes
FGFR1 amp and FGFR4 amp	2	FGFR1 amp and FGFR4 amp	yes	yes	yes
NTRK1 mut and PDGFRA mut	2	PDGFRA mut	yes	yes	yes
EPHA2 mut and FGFR2 mut	2	FGFR2 mut	yes	yes	yes
CSFR1 mut and EPHA2 over	2	EPHA2 over	yes	yes	yes
KDR mut and FRK over	2	FRK over	yes	yes	yes
FRK mut and FRK over	2	FRK over	yes	yes	yes
HIPK4 over and MUSK over	2	HIPK4 over and MUSK over	yes	no	no
DDR1 mut and SLK over	2	SLK over	yes	no	no
FGFR2 mut and PDGFRB mut	2	-	no	no	no
FLT1 mut and FLT4 mut	2	-	no	no	no
FLT4 mut and DDR2 mut	2	-	no	no	no
KDR mut and KIT mut	2	-	no	no	no
FLT4 mut and TNIK mut	2	-	no	no	no
MKNK2 mut and MAP3K7 mut	2	-	no	no	no
MAP4K4 mut	1	-	no	no	no
PDGFRA over	1	PDGFRA over	yes	yes	yes
RAF1 amp	1	RAF1 amp	yes	yes	yes
PDGFRB amp	1	PDGFRB amp	yes	yes	yes
BRAF mut	1	BRAF mut	yes	yes	yes
MAPK9 over	1	MAPK9 over	yes	no	yes
MAPK9 amp	1	MAPK9 amp	yes	no	yes
FRK over	1	FRK over	yes	yes	yes
HIPK4 over	1	HIPK4 over	yes	no	no
CDK8 amp	1	CDK8 amp	yes	no	no
EPHA2 mut	1	-	no	no	no
DDR2 mut	1	-	no	no	no
FGFR1 mut	1	-	no	no	no
FGFR3 mut	1		no	no	no
FLT1 mut	1	-	no	no	no
KIT mut	1	-	no	no	no
PDGFRA mut	1	-	no	no	no
TEK mut	1	-	no	no	no
TEK mut	1	-	no	no	no
TEK mut	1	-	no	no	no
RET mut	1	-	no	no	no
TEK mut	1	-	no	no	no
TIE1 mut	1	-	no	no	no
CSFR1 mut	1	-	no	no	no
EPHA6 mut	1	-	no	no	no
MKNK2 mut	1	-	no	no	no
CSFR1 mut	1	-	no	no	no
DDR1 mut	1	-	no	no	no
EPHA8 mut	1	-	no	no	no
MAP3K19 mut	1	-	no	no	no
MAP3K19 mut	1	-	no	no	no
MAP3K19 mut	1	-	no	no	no
MAPK14 mut	1	-	no	no	no
MAPK15 mut	1	-	no	no	no
MAPK15 mut	1	-	no	no	no
MAPK9 mut	1	-	no	no	no
TNIK mut	1	-	no	no	no
TNIK mut	1	-	no	no	no

mut, mutation; amp, amplification; over, overexpression; VUS, variant of uncertain significance; NMD, nonsense-mediated decay. ^a^ excluding benign and likely benign mutations.

**Table 3 cancers-17-00375-t003:** Results of three separate analyses to determine the presence of potential regorafenib targets in GBM patient samples. The presence of a potential regorafenib target was calculated in the extended 46-gene set (based on studies by Zopf et al. [14] and by Wilhelm et al. [15]), in the limited 18-gene set (based on data from DrugBank, PubChem, the EMA, and the FDA), and in a 23-gene set (comprising the 18 genes plus 5 angiogenesis-related genes not included in the 18-gene set).

Patients Included in Analysis	N Patients	N Genes Included in Analysis		
		46 Genes ^a^	18 Genes ^b^	23 Genes ^c^
		Patients with Potential Regorafenib Targets		
Patients with ≥1 alteration of any kind, excluding benign or likely benign mutations	73	73 (70.8%)	48 (46.6%)	49 (47.5%)
Patients with ≥1 putative oncogenic alteration	34	34 (33%)	26 (25.2%)	30 (29.1%)

^a^ The 46 genes encoding PKs identified by Zopf et al. [14] and Wilhelm et al. [15]; ^b^ the 18 genes encoding PKs inhibited at clinically achievable concentrations of regorafenib; ^c^ the 23 genes encoding PKs inhibited at clinically achievable concentrations of regorafenib (14 of which are also involved in angiogenesis) plus 5 additional genes involved in angiogenesis that do not encode PKs inhibited at clinically achievable concentrations of regorafenib.

## Data Availability

Molecular data underlying the findings described in the manuscript are fully available without restriction from the Bioproject Sequence Read Archive (RRID:SCR_004891): (https://www.ncbi.nlm.nih.gov/sra/PRJNA833243; https://www.ncbi.nlm.nih.gov/bioproject/PRJNA613395; https://www.ncbi.nlm.nih.gov/bioproject/PRJNA1073422) accessed on 15 January 2025.

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
