# Peer review of "Multikinase Treatment of Glioblastoma: Evaluating the Rationale for Regorafenib"

_cancers, 2025, doi:10.3390/cancers17030375_

Round 1

Reviewer 1 Report

Comments and Suggestions for Authors

Dear authors, 

I read your manuscript with interest. It is well-written overall and aims to evaluate if regorafenib can effectively treat glioblastoma.

I have some comments:

1) Results: Please provide a table with the characteristics of the included patients (at least age, sex). Moreover, age is really important since the 2021 WHO classification (WHO CNS5) classified pediatric glioblastoma differently from adult glioblastoma. These tumors were classified differently given their well-established molecular genetic differences. This aspect was not reported in the manuscript and needs to be taken into account.

2) The discussion of the manuscript can benefit from a wider overview of the efficacy and the safety of regorafenib (in both adult and pediatric patients) in patients with such disease (if available please search for meta-analysis) 

Author Response

Reviewer 1 questions

Authors’ responses

I read your manuscript with interest. It is well-written overall and aims to evaluate if regorafenib can effectively treat glioblastoma.

Thank you for your overall evaluation of the document. We appreciate the time you dedicated to this review and value your feedback.

I have some comments:

  1. Results: Please provide a table with the characteristics of the included patients (at least age, sex). Moreover, age is really important since the 2021 WHO classification (WHO CNS5) classified pediatric glioblastoma differently from adult glioblastoma. These tumors were classified differently given their well-established molecular genetic differences. This aspect was not reported in the manuscript and needs to be taken into account.

The reviewer is true. We added the supplementary table  S1 with patient characteristics. This acknowledges the reviewer’s suggestion by emphasizing the added value of including patient characteristics for context and understanding, even if it doesn’t alter the study's conclusions.

In addition, we added this paragraph to results:

‘The characteristics of the patients included in this study are detailed in Supplementary Table S1. Histological review confirmed a diagnosis of glioblastoma based on the WHO 2016 classification. According to the updated WHO 2021 classification, three patients would now be categorized as grade 4 astrocytoma due to the presence of an IDH1 mutation. None of the patients exhibited alterations in H3.3 genes. All participants underwent radiation therapy combined with concomitant and adjuvant temozolomide treatment.’

  1. The discussion of the manuscript can benefit from a wider overview of the efficacy and the safety of regorafenib (in both adult and pediatric patients) in patients with such disease (if available please search for meta-analysis).

The efficacy of regorafenib reported in the phase II REGOMA study is presented in the introduction. Regarding toxicity and tolerance, which we also evaluated, we observed that the toxicity reported in the REGOMA study was somewhat lower than that described for the drug in other diseases such as GIST, colon cancer, or hepatocellular carcinoma (hand-foot syndrome: 32% vs. 47-56.1%; diarrhea: 17% vs. 34-41%, among others)1-4. Since this topic is subject to multiple interpretations, we chose not to address the possible explanations, including pharmacological interactions. We have added to the article that the toxicity and tolerance were similar to those reported in other studies conducted in different diseases.

On the other hand, no meta-analyses have yet been published on the efficacy of regorafenib in glioblastoma, as the negative data from the AGILE trial were published in abstract format at the SNO congress in 2023. However, we have included a review on the topic published in 2024 by Mongiardi et al.,5  which examines alternative molecular effects on autophagy and stem cells that could explain the efficacy of regorafenib in glioblastoma. Nonetheless, since efficacy has not been demonstrated, this review could remain mere speculation.

Despite this, we have included this reference to this article in the discussion, (as a study limitation) as these mechanisms cannot be evaluated with the molecular studies we have used and would have been of interest had regorafenib proven useful in glioblastoma.

  1. Demetri GD, Reichardt P, Kang YK, et al. Efficacy and safety of regorafenib for advanced gastrointestinal stromal tumours after failure of imatinib and sunitinib (GRID): an international, multicentre, randomised, placebo-controlled, phase 3 trial. Lancet. 2013; 381(9863):295-302.
  2. Grothey A, Van Cutsem E, Sobrero A, et al. Regorafenib monotherapy for previously treated metastatic colorectal cancer (CORRECT): an international, multicentre, randomised, placebo-controlled, phase 3 trial. Lancet. 2013; 381(9863):303-312.
  3. Bruix J, Qin S, Merle P, et al. Regorafenib for patients with hepatocellular carcinoma who progressed on sorafenib treatment (RESORCE): a randomised, double-blind, placebo-controlled, phase 3 trial. Lancet. 2017; 389(10064):56-66.
  4. Lombardi G, De Salvo GL, Brandes AA, et al. Regorafenib compared with lomustine in patients with relapsed glioblastoma (REGOMA): a multicentre, open-label, randomised, controlled, phase 2 trial. Lancet Oncol. 2019; 20(1):110-119.
  5. Mongiardi MP, Pallini R, D'Alessandris QG, Levi A, Falchetti ML. Regorafenib and glioblastoma: a literature review of preclinical studies, molecular mechanisms and clinical effectiveness. Expert Rev Mol Med. 2024; 26:e5.

Reviewer 2 Report

Comments and Suggestions for Authors

High-grade Gliomas , and in particular Glioblastoma

are refractory to virtually all treatments.

The Blood-Brain Barrier BBB restrict the entry of all commonly used a

agents. If The BBB does not exist to the GBM 

then we can have a cure? can we increasing the MOs

if we treat animals in the flanc with induced GBM 

what ll be the outcome?

Author Response

Reviewer 2 questions

Authors’ responses

High-grade Gliomas , and in particular Glioblastoma

are refractory to virtually all treatments.

The Blood-Brain Barrier BBB restrict the entry of all commonly used a

agents. If The BBB does not exist to the GBM

then we can have a cure? can we increasing the MOs

if we treat animals in the flanc with induced GBM

what ll be the outcome?

Thank you for your overall evaluation of the document. We appreciate the time you dedicated to this review and value your feedback.

The reviewer's question is quite extensive and would require a separate article to address properly. To summarize: the blood-brain barrier (BBB) has indeed always been considered an obstacle to achieving sufficient concentrations of drugs in the tumor after metabolism, if necessary, to produce a therapeutic effect. For this, drugs must meet specific pharmacological and pharmacokinetic characteristics. While challenging, this is not impossible, as the design of many current drugs already considers this factor. For instance, vorasidenib was specifically designed for this purpose; it is an anti-IDH drug that crosses the BBB and has achieved spectacular results in controlling IDH-mutated gliomas. Of course, these drugs must not only cross the BBB but also avoid causing cerebral toxicity, as this is the very function of the BBB. Another issue would be the potential pharmacological interactions between drugs used in gliomas (such as corticosteroids and antiepileptics) that can significantly affect the metabolism of other medications, potentially reducing drug levels in the tumor.

In addition, it is true that the blood-brain barrier (BBB) is often disrupted around the tumor and metastases. However, the infiltrative portion of the tumors tends to retain an intact BBB, which further complicates achieving optimal drug concentrations in the tumor.

As a result, studies and trials in xenograft models have generally not translated into proven efficacy in humans. Currently, phase 0 studies are underway to evaluate drug concentrations in the tumor after prior administration before surgery. There are multiple factors that cannot be addressed in this comment, but in our opinion, neuro-oncology is currently on the right path to start achieving meaningful results for this awful disease.

Reviewer 3 Report

Comments and Suggestions for Authors

This study aimed to explore the the rationale of using regorafenib for the treatment of glioblastoma (GBM) based on the assessment of mutations, copy number variants [CNVs], fusions and overexpression in 46 genes encoding protein kinases (PKs) potentially targeted by regorafenib or its metabolites. Of the 46 genes,  a subset of 18 genes could be inhibited by regorafenib at clinically achievable concentrations.  Those 18 genes were present in 46.6% of the recruited patients (n = 103) and putative oncogenic alterations were present in only 25.2% of the patients. Given that only around 33% of patients with GBM had oncogenic alterations in any of the 46 potential targets of regorafenib, the benefits of regorafenib in the treatment of GBM are limited. 

Major comments:

1. Patient characteristics and demographic data were missing. 

2. In Figure 1 and Table 1, the 18 genes and associated proteins that can be inhibited by regorafenib at clinically achievable concentrations should be somehow highlighted. 

3. The speculation of poor regorafenib distribution in brain tumors should be taken with a grain of salt. Were there any in vivo or clinical studies demonstrating that the distribution of regorafenib in brain tumors (not in CSF) was poor? Drug distribution in brain tumors is supposed to be better than that in normal brain. Brain tumor cells often grow along the existing blood vessels, leading to a breakdown in blood brain barrier (BBB). Moreover, the neovasculature in brain tumors is usually leaky, which may result in an increase in drug penetration. 

4. The authors should demonstrate the association between the mutation status of any of the 18 genes and certain important clinicopathologic features of the GBMs. 

Author Response

Reviewer 3 questions

Authors’ responses

 This study aimed to explore the the rationale of using regorafenib for the treatment of glioblastoma (GBM) based on the assessment of mutations, copy number variants [CNVs], fusions and overexpression in 46 genes encoding protein kinases (PKs) potentially targeted by regorafenib or its metabolites. Of the 46 genes, a subset of 18 genes could be inhibited by regorafenib at clinically achievable concentrations. Those 18 genes were present in 46.6% of the recruited patients (n = 103) and putative oncogenic alterations were present in only 25.2% of

the patients. Given that only around 33% of patients with GBM had oncogenic alterations in any of the 46 potential targets of regorafenib, the benefits of regorafenib in the treatment of GBM are limited.

Thank you for the time you dedicated to this review and value your feedback.

Major comments:

  1. Patient characteristics and demographic data were missing.

The reviewer is true. We added the supplementary table  S1 with patient characteristics. This acknowledges the reviewer’s suggestion by emphasizing the added value of including patient characteristics for context and understanding, even if it doesn’t alter the study's conclusions.

In addition we added this paragraph to results:

‘The characteristics of the patients included in this study are detailed in Supplementary Table S1. Histological review confirmed a diagnosis of glioblastoma based on the WHO 2016 classification. According to the updated WHO 2021 classification, three patients would now be categorized as grade 4 astrocytomas due to the presence of an IDH1 mutation. None of the patients exhibited alterations in H3.3 genes. All participants underwent radiation therapy combined with concomitant and adjuvant temozolomide treatment.’

  1. In Figure 1 and Table 1, the 18 genes and associated proteins that can be inhibited by regorafenib at clinically achievable concentrations should be somehow highlighted.

At the reviewer’s suggestion, we have shaded in Table 1 the genes encoding proteins inhibited by regorafenib at clinically achievable concentrations. We have also done the same in Figure 1.

  1. The speculation of poor regorafenib distribution in brain tumors should be taken with a grain of salt. Were there any in vivo or clinical studies demonstrating that the distribution of regorafenib in brain tumors (not in CSF) was poor? Drug distribution in brain tumors is supposed to be better than that in normal brain. Brain tumor cells often grow along the existing blood vessels, leading to a breakdown in blood brain barrier (BBB). Moreover, the neovasculature in brain tumors is usually leaky, which may result in an increase in drug penetration.

We reviewed the penetration of regorafenib in the brain. There are no data from phase 0 studies detecting the drug concentration in the tumor when administered prior to surgical intervention. Some conflicting data exist. For example, the ADMET prediction (https://go.drugbank.com/drugs/DB08896#BE0000029) suggests a 0.851 probability of crossing the BBB. However, regorafenib and its metabolites exhibit high protein binding (99.5%), and BBB penetration is restricted by breast cancer resistance protein (BCRP/ABCG2) and P-glycoprotein (P-GP/ABCB1). In silico models predict poor BBB penetration of regorafenib (reference 38). Two in vivo studies report drug detection in the cerebrospinal fluid (CSF) significantly lower than that in serum (references 39 and 40 in the article). In an intact BBB, such as in the infiltrative but non-gadolinium-enhancing part of the tumor (brain adjacent to the tumor), penetration is likely suboptimal. Since no phase 0 study exists, we must rely on studies of CSF, which do not always represent the concentrations reached in the tumor but at least serve as a reference.

  1. The authors should demonstrate the association between the mutation status of any of the 18 genes and certain important clinicopathologic features of the GBMs.

Thank you for your suggestion, which we have taken into account. Our study is ongoing. We are participating in the RESPOND group, which examines clinical, radiological, and molecular factors to associate molecular alterations with specific clinicopathological features. Therefore, we reserve the studies suggested by the reviewer to be analyzed within the context of larger databases.1

  1. Akbari H, Bakas S, Sako C, et al. Machine Learning-based Prognostic Subgrouping of Glioblastoma: A Multi-center Study. Neuro Oncol. 2024.

Round 2

Reviewer 3 Report

Comments and Suggestions for Authors

The authors have addressed my concerns.